# Vitamin D Levels in Early and Middle Pregnancy and Preeclampsia, a Systematic Review and Meta-Analysis

**DOI:** 10.3390/nu14050999

**Published:** 2022-02-27

**Authors:** Kai-Lun Hu, Chun-Xi Zhang, Panpan Chen, Dan Zhang, Sarah Hunt

**Affiliations:** 1Key Laboratory of Reproductive Genetics (Ministry of Education), Department of Reproductive Endocrinology, Women’s Hospital, Zhejiang University School of Medicine, Hangzhou 310006, China; hukailun@bjmu.edu.cn (K.-L.H.); zhangcxi@zju.edu.cn (C.-X.Z.); 12018523@zju.edu.cn (P.C.); 2Key Laboratory of Women’s Reproductive Health of Zhejiang Province, Zhejiang University, Hangzhou 310006, China; 3Department of Obstetrics and Gynaecology, Monash University, Clayton, VIC 3168, Australia; sarah_prema@hotmail.com

**Keywords:** vitamin D, preeclampsia, pregnancy, systematic review, meta-analysis

## Abstract

Vitamin D (VitD) shows a beneficial role in placentation, the immune system, and angiogenesis, and thus, VitD status may link to the risk of preeclampsia. A meta-analysis was conducted to investigate the association between VitD status in early and middle pregnancy and the risk of preeclampsia. A total of 22 studies with 25,530 participants were included for analysis. Women with VitD insufficiency or deficiency had a higher preeclampsia rate compared to women with replete VitD levels (OR 1.58, 95% CI 1.39–1.79). Women with VitD deficiency had a higher preeclampsia rate compared to women with replete or insufficient VitD levels (OR 1.35, 95% CI 1.10–1.66). Women with insufficient VitD levels had a higher preeclampsia rate compared to women with replete VitD levels (OR 1.44, 95% CI 1.24–1.66). Women with deficient VitD levels had a higher preeclampsia rate compared to women with replete VitD levels (OR 1.50, 95% CI 1.05–2.14). Sensitivity analysis showed the results were stable after excluding any one of the included studies. In conclusion, our systematic review suggested that VitD insufficiency or deficiency was associated with an increased risk of preeclampsia.

## 1. Introduction

Preeclampsia is a multisystem disease during pregnancy, characterized by the new-onset of gestational hypertension and proteinuria. It occurs in around 3–8% of all pregnancies and is associated with increased maternal and fetal morbidity and mortality [1,2]. Maternal preeclampsia is also associated with a higher incidence of cardiovascular and kidney disease in the later life of the child [3,4,5]. Currently, delivery is the only curative therapy for preeclampsia, and pharmacological management is symptomatic treatment only. Therapies aimed at preventing preeclampsia therefore are a priority for ongoing investigation.

VitD deficiency is common during pregnancy, with a prevalence ranging from 8–70% depending on skin pigmentation and sunlight exposure [6,7,8,9]. Accumulating evidence suggests that VitD deficiency may be implicated in recurrent pregnancy loss, adverse obstetrical and neonatal outcomes [10,11]. More recently, attention has been paid to the potential association between VitD levels in pregnancy and the risk of preeclampsia. It has been postulated that increased VitD levels may improve the invasion of the human extravillous trophoblast, which is required for normal placentation [12]. Additionally, accumulating evidence suggested that VitD has a beneficial effect on endothelial repair and angiogenesis [13,14,15] and that VitD deficiency is associated with the pathogenesis of cardiovascular diseases and arterial hypertension [16,17]. Therefore, it is likely that VitD has a role in improving endothelial repair and angiogenesis and controlling blood pressure in preeclampsia [18]. Furthermore, the immunomodulatory properties of VitD may also reduce the risk of preeclampsia development [13,19,20]. Indeed, previous observational studies have demonstrated significant association between VitD deficiency and increased preeclampsia risk [21,22,23], whereas others suggest no association between maternal VitD deficiency and preeclampsia rate [11,24,25,26]. Overall, the epidemiological evidence from observational data is conflicting and most studies are limited by small sample sizes. Additionally, there is a lack of consensus regarding definition of VitD deficiency and the potential threshold level associated with preeclampsia risk. Although there have been several systematic review and meta-analysis papers related to the association of VitD with preeclampsia [27,28,29,30], some important studies were not included in these meta-analyses. Additionally, most studies included in these systematic reviews were case-control studies that investigated women in late pregnancy. Even when the association between VitD deficiency in late pregnancy and obstetric complications is detected, the prevention of these diseases by the modification of VitD status is too late.

In this systematic review, we aimed to determine the association between VitD levels in early or middle pregnancy (≤24 weeks) and the risk of pre-eclampsia.

## 2. Methods

### 2.1. Eligibility Criteria

The protocol of this systematic review was prospectively registered in PROSPERO (reference: CRD42021271154). We reported this systematic review according to the guideline of The Preferred Reporting Items for Systematic reviews and Meta-Analyses (PRISMA) statement 2020 [31]. We included all studies that investigated the association of VitD levels with the rate of preeclampsia. Because the detection of the association between VitD deficiency in early or middle pregnancy (rather than late pregnancy) and later obstetric complications may allow the prevention of these diseases by screening and modification of VitD status in early or middle pregnancy [24], we excluded studies if they investigated pregnant women over 24 gestational weeks. Reviews, case report studies, and study protocols were also excluded.

### 2.2. Search Strategy

Two authors (KLH and CXZ) independently searched the database of PubMed, EMBASE, Cochrane library, and Web of Science from January 1990 to July 2021. The search terms included “preeclampsia”, “vitamin D”, and “hypertensive disorder of pregnancy”. The detailed search terms could be seen in Appendix A.

### 2.3. Selection Process

Two authors (KLH and CXZ) independently reviewed the titles and abstracts based on the predefined eligibility criteria. The full manuscripts were obtained when the titles and abstracts were considered to be related. Any disagreement between the two authors was resolved by a third review author. References from all included studies were checked to identify relevant articles not captured by the electronic searches.

### 2.4. Risk of Bias Assessment

Two reviewers (KLH and CXZ) independently assessed the quality of the included studies. Cohort and case-control studies were assessed according to The Newcastle-Ottawa Quality Assessment Scales [32]. Cross-sectional studies were assessed using the Agency for Healthcare Research and Quality (AHRQ).

### 2.5. Data Collection Process

Two reviewers (KLH and CXZ) independently extracted the data from included studies. If a study with multiple publications was found, the main report was used as the reference with additional details supplemented from other papers.

VitD levels were stratified into three groups according to Endocrine Society recommendations [9]: the replete level (>30 ng/mL or >75 nmol/L), the insufficient level (20–30 ng/mL or 50–75 nmol/L), and the deficient level (<20 ng/mL or <50 nmol/L). A few studies used 15 ng/mL (37.5 nmol/L) as the cutoff, and thus <15 ng/mL was considered as the deficient level in these studies. The primary outcome was preeclampsia, defined by the new-onset of gestational hypertension and proteinuria.

### 2.6. Synthesis Methods

The Stata 15.0 (StataCorp, College Station, TX, USA) was as used to perform the meta-analysis using either the inverse-variance weighted model (fixed-effect) or the DerSimonian and Laird model (random-effect). Both were displayed in the forest plot, and the fixed-effect model was applied if no significant heterogeneity was identified (I^2^ < 50%), whereas a random-effect model was used a significant heterogeneity was detected (I^2^ > 50%),. The combined data was shown in a pooled odds ratio (OR) with a 95% confidence interval (CI). Publication bias was assessed by funnel plot asymmetry as well as Egger’s test. Sensitivity analysis was conducted by omitting each individual study in turn to explore the effect of a single study on the overall meta-analysis. Subgroup analysis was conducted according to the gestational weeks of pregnancy (early ≤ 14 weeks, middle 15–24 weeks, early and middle not specified) and study design (case-control, cohort, cross-sectional). Statistical significance was set at α equals to 0.05.

## 3. Results

### 3.1. Characteristics of the Included Studies

Diagramatic representation of the review process is outlined in Appendix A. A total of 22 studies with a sample size of 25,530 were included for analysis [11,21,22,24,25,26,33,34,35,36,37,38,39,40,41,42,43,44,45,46,47,48]. The characteristics of the included studies are shown in Table 1. Study quality assessment were shown in Appendix A (case-control studies), Appendix A (cohort studies), and Appendix A (cross-sectional studies).

The included studies varied in publication date from 2007 to 2020. Eleven were case-control studies; eight were cohort studies and three were cross-sectional studies (Table 1). Sample sizes varied from 142 women to 5109 women. Six studies focused on women in early pregnancy and three studies included women in middle pregnancy (Table 1). While most studies defined insufficiency and deficiency at 30 ng/mL and 20 ng/mL, respectively, four studies used 15 ng/mL as the diagnostic cutoff [11,25,34,35] (Appendix A). Nearly all studies used multivariable analysis; however, the number and type of potential confounders controlled for in the final analyses was not uniform between studies (Appendix A).

### 3.2. Replete Levels of VitD (≥30 ng/mL) versus Insufficient or Deficient Levels of VitD (<30 ng/mL)

A total of 17,719 pregnant women (*n* = 12,908 with replete levels of VitD and *n* = 4811 with insufficient or deficient levels of VitD) from 14 studies were included. Women with VitD < 30 ng/mL showed an increased preeclampsia rate compared to women with VitD ≥ 30 ng/mL (OR 1.58, 95% CI 1.39–1.79, I^2^ = 34%, fixed-effect) (Figure 1). No publication bias was detected (Appendix A, Egger’s test: *p* = 0.069). Sensitivity analysis demonstrated the results were stable after excluding any one of the included studies (Appendix A). There was a trend toward an increased risk of preeclampsia in women with VitD < 30 ng/mL in both early pregnancy and middle pregnancy, but they were not statistically significant (early pregnany: OR 1.29, 95% CI 0.93–1.79, I^2^ = 0%, fixed-effect; middle pregnancy: OR 1.41, 95% CI 0.97–2.06, I^2^ = 18%, fixed-effect) (Figure 1a). The pooled data from studies with a case-control design showed a significantly higher risk of preeclampsia in women with VitD < 30 ng/mL, but it was not seen in pooled data from cohort studies or cross-sectional studies (case-control: OR 1.80, 95% CI 1.56–2.09, I^2^ = 0%, fixed-effect; cohort: OR 1.09, 95% CI 0.85–1.40, I2 = 0%, fixed-effect; cross-sectional: OR 0.76, 95% CI 0.18–3.19, I^2^ = 0%, fixed-effect) (Figure 1b).

### 3.3. Replete or Insufficient Levels of VitD (≥20 ng/mL) versus Deficient Levels of VitD (<20 ng/mL or < 15 ng/mL)

A total of 23,217 pregnant women (*n* = 8084 with replete levels of VitD and *n* = 15,133 with insufficient or deficient levels of VitD) from 19 studies were included in this analysis. Women with VitD < 20 ng/mL had a higher preeclampsia rate as compared to women with VitD ≥ 20 ng/mL (OR 1.35, 95% CI 1.10–1.66, I^2^ = 57%, random-effect) (Figure 2). No publication bias was detected (Appendix A, Egger’s test: *p* = 0.415). Sensitivity analysis demonstrated the results were stable after excluding any one of the included studies (Appendix A). There was a trend toward an increased risk of preeclampsia in women with VitD < 20 ng/mL in middle pregnancy but no significant difference in preeclampsia risk was seen in early pregnancy (early pregnany: OR 0.85, 95% CI 0.62–1.18, I^2^ = 19%, fixed-effect; middle pregnancy: OR 1.59, 95% CI 0.87–2.92, I^2^ = 61%, random-effect) (Figure 2a). The pooled data from studies with a case-control or cohort design demonstrated significantly increased preeclampsia rates in association with VitD < 20 ng/Ml, but it was not seen in pooled data from cross-sectional studies (case-control: OR 1.46, 95% CI 1.02–2.10, I^2^ = 77%, random-effect; cohort: OR 1.24, 95% CI 1.02–1.51, I2 = 0%, fixed-effect; cross-sectional: OR 1.08, 95% CI 0.49–2.35, I^2^ = 0%, fixed-effect) (Figure 2b).

### 3.4. Replete Levels of VitD (≥30 ng/mL) versus Insufficient Levels of VitD (20–30 ng/mL or 15–30 ng/mL)

A total of 11,091 pregnant women (*n* = 6856 with replete levels of VitD and *n* = 4235 with insufficient or deficient levels of VitD) from 11 studies were included in this analysis. Women with VitD insufficiency were more likely to develop preeclampsia as compared to women who were VitD replete (OR 1.44, 95% CI 1.24–1.66, I^2^ = 23%, fixed-effect) (Figure 3). Publication bias was detected (Appendix A, Egger’s test: *p* = 0.005). Sensitivity analysis showed the results were stable after excluding any one of the included studies (Appendix A). A slightly increased risk of preeclampsia was seen in women with insufficient VitD levels in early pregnancy, but it was not seen in middle pregnancy (early pregnancy: OR 1.40, 95% CI 1.00–1.96, I^2^ = 22%, fixed-effect; middle pregnancy: OR 1.21, 95% CI 0.83–1.76, I^2^ = 0%, fixed-effect) (Figure 3a). The pooled data from studies with a case-control design showed a significantly higher risk of preeclampsia in women with insufficient VitD levels, but it was not seen in pooled data from cohort studies or cross-sectional studies (case-control: OR 1.59, 95% CI 1.35–1.88, I2 = 0%, fixed-effect; cohort: OR 1.02, 95% CI 0.75–1.39, I^2^ = 11%, fixed-effect; cross-sectional: OR 0.85, 95% CI 0.14–5.19) (Figure 3b).

### 3.5. Replete Levels of VitD (≥30 ng/mL) versus Deficient Levels of VitD (<20 ng/mL or <15 ng/mL)

A total of 8550 pregnant women (*n* = 4315 with replete levels of VitD and *n* = 4235 with deficient levels of VitD) from 11 studies were included in this analysis. Women with VitD deficiency were more likely to develop preeclampsia as compared to women with replete VitD levels (OR 1.50, 95% CI 1.05–2.14, I^2^ = 64%, random-effect) (Figure 4). No publication bias was detected (Appendix A, Egger’s test: *p* = 0.188). Sensitivity analysis showed the results were stable after excluding any one of the included studies (Appendix A). A slightly increased risk of preeclampsia was seen in women with deficient VitD levels in middle pregnancy, but it was not seen in early pregnancy (early pregnancy: OR 0.96, 95% CI 0.61–1.51, I^2^ = 0%, fixed-effect; middle pregnancy: OR 1.78, 95% CI 0.93–3.41, I^2^ = 54%, random-effect) (Figure 4a). The pooled data from studies with a case-control design showed a significantly higher risk of preeclampsia in women with VitD deficiency, but it was not seen in pooled data from cohort studies or cross-sectional studies (case-control: OR 1.99, 95% CI 1.24–3.18, I^2^ = 71%, random-effect; cohort: OR 1.10, 95% CI 0.81–1.50, I2 = 0%, fixed-effect; cross-sectional: OR 0.71, 95% CI 0.14–3.62, I^2^ = 0%, fixed-effect) (Figure 4b).

## 4. Discussion

Our study included data from 22 observational studies and demonstrated an association between VitD deficiency or insufficiency and pre-eclampsia risks in early to middle pregnancy.

The pathogenesis of preeclampsia remains incompletely understood. It is proposed that incomplete remodeling of spiral arteries of the uterus during placentation induces limited perfusion and hypoxia of the placenta, with subsequent release of antiangiogenic factors into the maternal circulation [49,50,51]. These antiangiogenic factors, including tyrosine kinase-1 and soluble endoglin, can lead to endothelial damage and the clinical features that define preeclampsia [52,53]. In addition, a large amount of proinflammatory cytokines (IL-1β, IL-6, and IL-8) are released from the neutrophils and monocytes in the decidua, leading to increased damage to the blood vessels [54,55]. A previous study demonstrated that increased VitD levels improved the invasion of the human extravillous trophoblast and thus proposed that VitD may exert a preventive effect on the development of preeclamptic disease [12]. Additionally, accumulating evidence have suggested that VitD plays a beneficial role in endothelial repair and angiogenesis [13,14,15] and that VitD deficiency is associated with the pathogenesis of cardiovascular diseases and arterial hypertension [16,17]. Therefore, VitD may have a role in augmenting endothelial repair and angiogenesis and controlling blood pressure in preeclampsia [18] and its immunomodulatory properties may also reduce the risk of preeclampsia development [13,19,20].

The association between VitD deficiency during pregnancy and preeclampsia has been much investigated. Previous systematic reviews of observational studies have demonstrated that women with VitD deficiency (at cutoff 20 ng/mL) were more likely to develop preeclampsia [27,28,56,57]. However, the gestation at which the blood was collected are not fully discussed in these systematic reviews. In this review, we focused on the association of VitD levels in early and middle pregnancy, and VitD supplementation at these points in pregnancy may provide an opportunity for disease prevention or modification. Our review also included several new studies and important studies that have been omitted in previous meta-analyses. Moreover, most previous reviews have compared VitD levels ≥ 20 ng/mL with VitD levels < 20 ng/mL, whereas our study compared VitD levels ≥ 30 ng/mL, <30 ng/mL, and <20 ng/mL. Our study demonstrates that VitD levels higher than 30 ng/mL are associated with reduced preeclampsia risk when compared with VitD levels < 30 ng/mL or <20 ng/mL, suggesting that women may benefit from the supplementation of VitD to a level of ≥30 ng/mL in early and middle pregnancy. This has been supported by previous observational studies and meta-analysis which demonstrated that VitD supplementation during pregnancy was related to a reduced rate of preeclampsia [58,59,60,61,62]. However, a systematic review of randomized trials found VitD supplementation did not significantly alter preeclampsia risk [63]. It should be noted that only three randomized trials with a total sample size of 654 were included in this systematic review [63], which may explain the inconsistency with other studies. A previous randomized trial suggested that VitD supplementation (4400 vs. 400 IU/d) in 10–18 weeks of pregnancy was not able to reduce preeclampsia risks [24]. However, VitD ≥ 30 ng/mL at the trial entry were related a reduced rate of preeclampsia [24]. It should be noted that only 74% pregnant women has a replete VitD levels in the 4400 IU/d group in late pregnancy [24], which may explain the nonsignificant difference of preeclampsia incidence for VitD supplementation in early or middle pregnancy. Future trials should further investigate whether VitD supplementation to the replete levels during pregnancy is associated with reduced preeclampsia.

Our review has several strengths. We limited the gestational age of blood collection to early and middle pregnancy. Additionally, we compared replete, insufficient, and deficient VitD status and found that women who were VitD replete had lower rates of preclampsia than women with VitD insufficiency or deficiency. We also conducted sensitivity analysis and demonstrated that our results were robust after omitting any one of the included studies.

This systematic review also has several limitations. The available studies were heterogeneous in terms of gestational weeks, study design, methods of VitD measurement, and skin characteristics. Additionally, we included observational studies; therefore, there is the potential that unrecognized confounders may have impacted our results. Recognizing the limitations of studies included in meta-analyses, however, forms a basis for future studies with more optimal design and methods to define the role of VitD in preeclampsia. Future RCTs should consider initiating VitD supplementation in early or middle pregnancy or even before pregnancy in women with VitD insufficiency or deficiency.

In conclusion, our systematic review demonstrated that VitD insufficiency (20–30 ng/mL) or deficiency (<20 ng/mL) was associated with an increased risk of preeclampsia. This raises the possibility that VitD supplementation in early or middle pregnancy may represent a risk-modifying therapy.

## Figures and Tables

**Figure 1 nutrients-14-00999-f001:**
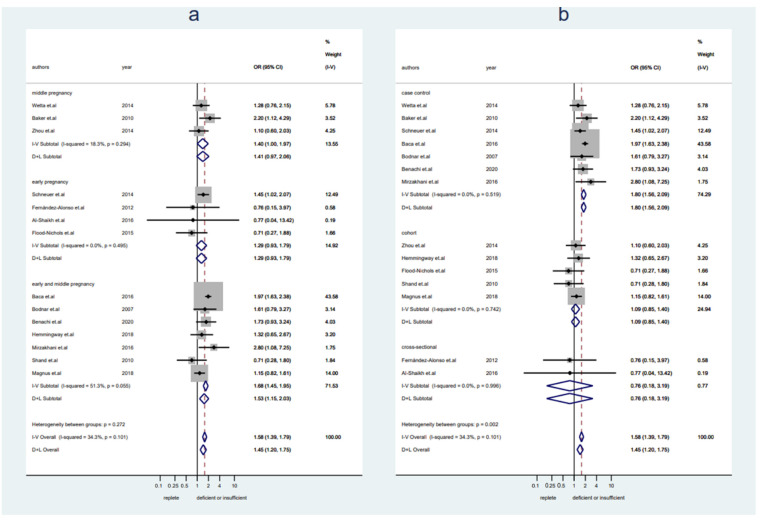
Studies evaluating replete vitamin D levels versus insufficient or deficient vitamin D levels. (**a**): Studies were stratified by gestational weeks (early, middle, and early and middle not specified); (**b**): studies were stratified by the study design.

**Figure 2 nutrients-14-00999-f002:**
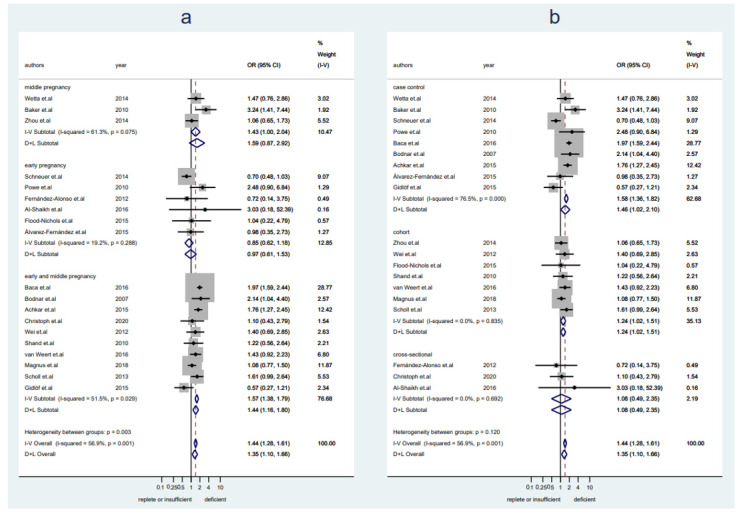
Studies evaluating replete vitamin D levels or insufficient versus deficient vitamin D levels. (**a**): Studies were stratified by gestational weeks (early, middle, and early and middle not specified); (**b**): studies were stratified by the study design.

**Figure 3 nutrients-14-00999-f003:**
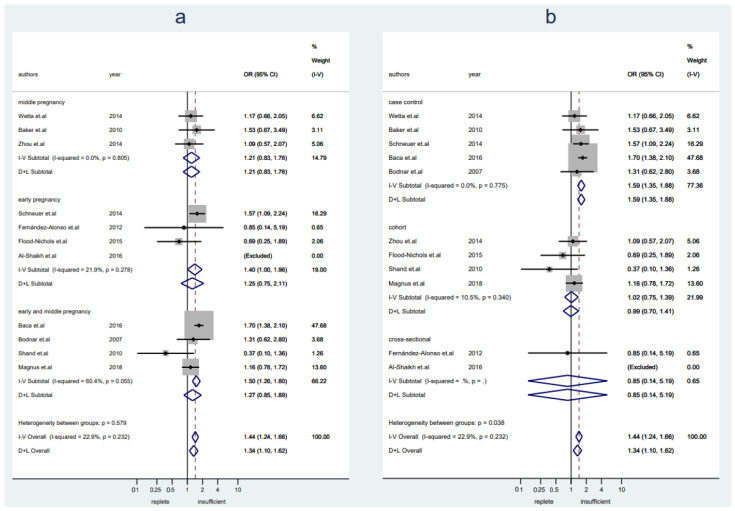
Studies evaluating replete vitamin D levels versus insufficient vitamin D levels. (**a**): Studies were stratified by gestational weeks (early, middle, and early and middle not specified); (**b**): studies were stratified by the study design.

**Figure 4 nutrients-14-00999-f004:**
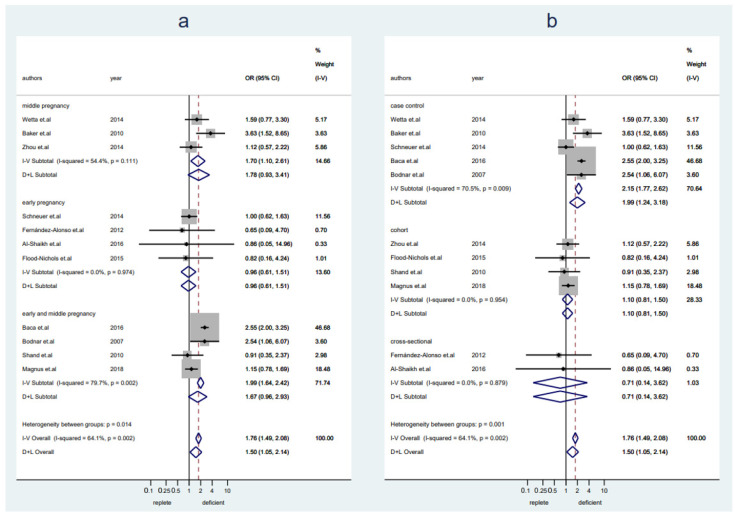
Studies evaluating replete vitamin D levels versus deficient vitamin D levels. (**a**): Studies were stratified by gestational weeks (early, middle, and early and middle not specified); (**b**): studies were stratified by the study design.

**Table 1 nutrients-14-00999-t001:** Characteristics of included studies.

Studies	Study Type	Eligibility for Pregnant Women	Method of Measurement	Gestational Weeks of Sampling	Sample Size	Location
[11]	Nested case–control	Women with multiple gestations, fetal anomalies, or maternal medical complications were excluded.	LCMS	15–21	266	Birmingham, America/52.3° N
[33]	Nested case–control	Women with multiple gestations, major congenital fetal anomalies, pregestational hypertension, kidney disease, diabetes mellitus, known thrombophilias, PCMD were excluded.	LCMS	15–20	241	Boston, America /52.58° N
[26]	Cohort	Women with abnormal liver function, chronic disease and tumor; severe infections, trauma or in perioperative, before 13 weeks of gestation, and women who take corticosteroids, drug abuse (including alcohol) were excluded.	ECLIA	16–20	1953	Guangzhou, China/23.1° N
[34]	Nested case–control	Women with adverse pregnancy outcomes were excluded	CLIA	10–14	5109	New South Wales, Australia /33.9°
[25]	Nested case–control	Women with GDM or give birth to SGA infants	LCMS	≤14	170	Boston, America /52.58° N
[22]	Nested case–control	Women who had aneuploidy screening at 20 weeks or less gestation and who subsequently delivered live born infants.	LCMS	≤20	2327	Pennsylvania, America/40.3°
[35]	Nested case–control	Nulliparous women aged 14–44 years, carrying singleton infants.	ELISA	≤22	265	Pennsylvania, America/40.3°
[36]	Nested case–control	Women with multiple gestations, calcium imbalance, hypertension, renal insufficiency, bone disease, lithium therapy, bowel malabsorption, or kidney stone disease were excluded.	RIA	≤15	402	six centers: one in Belgium and five in France
[21]	Nested case–control	Women with preexisting hypertension, missing essential outcome information (no gestational age at enrollment), or multiple gestations were excluded.	CLIA	≤20	2048	Quebec, Canada/46.5° N
[37]	Cross-sectional	Women with increased risks for intrauterine fetal growth restriction, hereditary thrombophilias, or acquired thrombophilias were excluded.	ECLIA	11–14	466	Almería, Spain/36.8°
[38]	Cross-sectional	NA	CLIA	≤24	1382	Bern, Switzerland/46.5°
[39]	Cross-sectional	Women with PCMD, metabolic bone disease, liver, kidney, or gastrointestinal diseases and the use of vitamin D supplements.	ELISA	≤12	1000	Saudi Arabia/24.3°
[40]	Cohort	Nulliparous women with a low-risk singleton pregnancy. Pregnancies at increased risk of pre-eclampsia, SGA, or spontaneous preterm birth or medical history, known major fetal anomaly or abnormal karyotype were excluded.	LCMS	<16	1754	Cork, Ireland/51.9° N
[24]	Nested case–control	Maternal age between 18 and 39 years and not a current smoker or a user of other nicotine products. Women with PCMD, multiple pregnancies, vitamin D taken (>2000 IU per day), fetal anomalies, or ART use were excluded.	CLIA	10–18	157	United States
[41]	Cohort	Women who regularly took 200 mg/d for vitamin C and/or 50 IU/d for vitamin E, or warfarin, or with fetal abnormalities, or with PCMD, or with repeated spontaneous abortion were excluded.	CLIA	12–18	697	Canada and Mexico
[42]	Cohort	Healthy, nulliparous women aged 18 years or older without PCMD or infertility treatment. Patients with predictors for hypovitaminosis D or a prior pregnancy that had progressed beyond the first trimester and resulted in a fetal loss were excluded.	ELISA	8–12	235	United States
[43]	Cohort	Women aged ≥ 18 years with either clinical or biochemical risk factors for pre-eclampsia	RIA	10–20	221	Canada/49° N
[44]	Cohort	Nulliparous women with a singleton pregnancy.	ELISA	<17	2074	Amsterdam, the Netherlands
[45]	Cohort	Gestational age < 24 weeks, resident in Rotterdam at the date of delivery, expected delivery date lies between June 2002 and July 2004	LCMS	<24	3323	Rotterdam, the Netherlands
[46]	Cohort	Healthy pregnant women. Gravidae with serious nonobstetric problems are not eligible.	LCMS	<20	1141	Camden, United States
[47]	Nested case–control	Suspected PE over 20 weeks of gestation between January 2010 and March 2013. Women who were diagnosed with PE before their presentation at the emergency department were not included.	CLIA	9–12	142	Oviedo, Spain
[48]	Nested case–control	After identifying women who developed preeclampsia, the control group was drawn by random selection and comprised 10 women delivered in each month of the year	CLIA	Mean (SD): 12 (3)	157	Malmo, Sweden /55°37′ N

Abbreviations: Premature rupture of membranes, PROM; preeclampsia, PE; gestational diabetes mellitus, GDM; small-for-gestational-age, SGA; diabetes mellitus, DM; preexisting chronic medical disease, PCMD; liquid chromatography–tandem mass spectrometry, LCMS; enzyme-linked immunosorbent assay, ELISA; electrochemiluminescence immunoassay, ECLIA; radioimmunoassay, RIA; chemiluminescent immunoassays, CLIA; assisted reproductive techniques, ART.

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
