# Peer review of "Vitamin D Levels in Early and Middle Pregnancy and Preeclampsia, a Systematic Review and Meta-Analysis"

_nutrients, 2022, doi:10.3390/nu14050999_

Round 1

Reviewer 1 Report

The focus of this meta-analysis is to determine the association between vitamin D levels in early or middle pregnancy (<=24 weeks) and the risk of developing preeclampsia. Several observational studies have been included in the analysis and results of meta-analysis are presented in detail. However, this report fails to provide comprehensive review of the topic at hand. Some suggestions and comments are presented below.

Introduction

  • Rationale of vitamin D need during pregnancy is lacking. Information regarding vitamin D metabolism during pregnancy can be included. Second paragraph of this review’s discussion can be mentioned in the introduction section to strengthen the rationale.

Methods

  • It is clear that studies included were with participants in early and middle pregnancy but rationale for excluding late state pregnancy is not mentioned at all.
  • It is not clear why only observational studies were included. If RCT can not be included in the meta-analysis, at least interventional RCT’s findings should be discussed.

Results

  • Results are not reported in a manner of systematic review at all. All the results include meta-analysis findings.
  • Table 1- Characteristics of included studies need to include more data e.g. inclusion/exclusion criteria needs clarification, maternal age, each study outcome.

Discussion

  • Discussion needs major revision. It is mostly re-reporting of results. It can be strengthened by reporting interpretation of results, providing suggestions of if and/or how information gained can be useful for future studies or in clinical decision making.
  • This meta analysis did not include any RCT or interventional studies. A comparison with results from interventional studies would help understand if this meta-analysis findings are supported or contradicted.

Author Response

The focus of this meta-analysis is to determine the association between vitamin D levels in early or middle pregnancy (<=24 weeks) and the risk of developing preeclampsia. Several observational studies have been included in the analysis and results of meta-analysis are presented in detail. However, this report fails to provide comprehensive review of the topic at hand. Some suggestions and comments are presented below.

 Response: we thank the reviewer for the positive comments. We have revised the manuscript according to the reviewer’s suggestions.

Introduction

  • Rationale of vitamin D need during pregnancy is lacking. Information regarding vitamin D metabolism during pregnancy can be included. Second paragraph of this review’s discussion can be mentioned in the introduction section to strengthen the rationale.

Response: we have mentioned the rationale of VD in the introduction section. Please see page 1-2, line 48-50

Methods

  • It is clear that studies included were with participants in early and middle pregnancy but rationale for excluding late state pregnancy is not mentioned at all.

Response: That is because the detection of the association between vitamin D deficiency in early or middle pregnancy (rather than late pregnancy) and later obstetric complications may allow the prevention of these disease by taking vitamin D. Another reason is that if we include all pregnancies, there are many low quality case-control studies that investigates women in late pregnancy. Please see page 2, line 60-66, 75-76

  • It is not clear why only observational studies were included. If RCT can not be included in the meta-analysis, at least interventional RCT’s findings should be discussed.

Response: our study aims to investigate the association between vitamin D levels and preeclampsia. There is no intervention and thus no RCT will be included. However, secondary analysis from RCTs that investigate the association between VD and preecalmsia will be included, such as Mirzakhani et al. We also discussed the RCTs in the discussion section. Please see page 10, line 274-283

Results

  • Results are not reported in a manner of systematic review at all. All the results include meta-analysis findings.
  • Table 1- Characteristics of included studies need to include more data e.g. inclusion/exclusion criteria needs clarification, maternal age, each study outcome.

Response: inclusion/exclusion criteria has been described in the eligibility col. Some studies just investigate preeclampsia, some with many outcomes, and some with gene sequencing outcome. The maternal age is described totally in different studies, some describe by control or case, some describe by vitamin d levels (sufficient, deficient), and some categorize maternal age (10-20, 20,30 40, 50). We insist that it is unnecessary to included maternal age and each study outcome in Table 1.

Discussion

  • Discussion needs major revision. It is mostly re-reporting of results. It can be strengthened by reporting interpretation of results, providing suggestions of if and/or how information gained can be useful for future studies or in clinical decision making.

Response: we thank the reviewer’s suggestions. we have included the interpretation of results that vitamin D may have a role in PE incidence. We also provide suggestions that “Future trials should further investigate whether vitamin D supplementation to the replete levels during pregnancy is associated with reduced preeclampsia.” Please see page 10 line 281-283

  • This meta analysis did not include any RCT or interventional studies. A comparison with results from interventional studies would help understand if this meta-analysis findings are supported or contradicted.

Response: we did discuss the findings from RCTs and have a short discussion. Please see page 10 line 268-280

Reviewer 2 Report

The manuscript is a systematic review and meta-analysis of the literature about the relationship between preeclampsia and hypovitaminosis D. The authors have applied commendably all the guidelines, reporting the search key, searching in more than 3 databases, and looking at the quality of the papers. Thus the methods are robust.  I have only few comments for the authors:

  • Table 1 might be implemented with quality evaluation, helping the readers
  • Have you evaluated if studies have included seasons in their analyses? Sun exposure has a key role in vitamin D and this aspect should be stressed
  • I think the authors could have a higher impact showing the next steps for research and future studies, using the results of their analysis
  • Please revised the figure because they are difficult to read 

Author Response

The manuscript is a systematic review and meta-analysis of the literature about the relationship between preeclampsia and hypovitaminosis D. The authors have applied commendably all the guidelines, reporting the search key, searching in more than 3 databases, and looking at the quality of the papers. Thus the methods are robust.  I have only few comments for the authors:

Response: we thank the reviewer for the positive comments. We have revised the manuscript according to the reviewer’s suggestions.

  • Table 1 might be implemented with quality evaluation, helping the readers

Response: the quality assessment of each study was shown in supplemental files.

  • Have you evaluated if studies have included seasons in their analyses? Sun exposure has a key role in vitamin D and this aspect should be stressed

Response: thanks for the suggestion. some studies have included the season of blood collection, while others were not. It has been listed in supplemental table 5

  • I think the authors could have a higher impact showing the next steps for research and future studies, using the results of their analysis

Response: thanks for the suggestion. We provide suggestions that “Future trials should further investigate whether vitamin D supplementation to the replete levels during pregnancy is associated with reduced preeclampsia.” Please see page 10 line 281-283

Please revised the figure because they are difficult to read 

Response: we have increased the dpi of all figures.

Round 2

Reviewer 1 Report

Thank you for addressing all the comments.